# Computing Planning Centroids and Minimum Covering States using Symbolic Bidirectional Search

## Submission #218

### Abstract

In some scenarios, planning agents might be interested in reaching states that keep certain relationships with respect to a set of goals. Recently, two of these types of relationships were proposed: centroids, which minimize the average distance to the goals; and minimum covering states, which minimize the maximum distance to the goals. Previous approaches compute these states by searching forward either in the original or a reformulated task. In this paper, we propose several algorithms that use symbolic bidirectional search to efficiently compute centroids and minimum covering states. Experimental results in existing and novel benchmarks show that our algorithms scale much better than previous approaches, establishing a new state-of-the-art technique for this problem.

## 1 Introduction

Automated Planning typically deals with the task of finding a sequence of actions, namely a plan, which achieves a goal state from a given initial state (Ghallab, Nau, and Traverso 2004). However, in some scenarios planning agents might be interested in reaching states that keep certain relationships with respect to a set of (potential) goals. Recently, two of these states were proposed (Pozanco et al. 2019; Karpas 2022): centroids, which minimize the average distance to the goals; and minimum covering states, which minimize the maximum distance to the goals. These states have proven to be useful for tasks such as deceptive planning (Price et al. 2023) or anticipatory planning (Burns et al. 2012).

Figure 1 illustrates the centroid (blue) and minimum covering states (green) of the planning task introduced by Pozanco et al. (2019), where a forest ranger has to put out fires that might arrive dynamically in the locations marked with a flame. Under these circumstances, the ranger should generate a plan to set the camp at a location that minimizes the cost (time) of a plan to put out any fire that might arrive.

There exist two approaches in the literature to compute these states. The first one, introduced by Pozanco et al. (2019), uses a best-first search algorithm that expands all the reachable states from the initial state, computing one

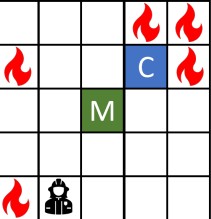

Figure 1: Illustration of the centroid (blue) and minimum covering (green) states of a forest ranger planning task. Flames mark locations where the forest ranger might need to put out fires (goals).

optimal plan to each goal from each state. The second approach, introduced by Karpas (2022), compiles the original planning task into a multi-agent planning task where each agent is trying to achieve one of the goals. They proposed two slightly different compilations for centroids and minimum covering states, both being inspired by seminal work on goal recognition design (Keren, Gal, and Karpas 2014, 2015). In the reformulated task to compute centroids, agents can either perform actions together, with an associated cost of 0, or split and perform actions separately, with their original cost. The idea behind the compilation is that the agents will try to execute as many joint actions as possible, and the state where they start executing separate actions will be the centroid of the planning task. Empirically, Karpas (2022) showed that computing centroid and minimum covering states by solving the reformulated tasks was orders of magnitude faster than using the exhaustive search approach by Pozanco et al. (2019) in most planning tasks.

In this paper we propose to use symbolic bidirectional search to compute centroid and minimum covering states. We believe this search paradigm offers two main benefits over previous approaches. First, symbolic search uses succint data structures to efficiently represent and manipulate sets of states. This allows us to compute all the centroid and minimum covering states without conducting exhaustive search. Second, these states are defined as those that (i) optimize a given statistical measure with respect to their distance to the goals; and (ii) are reachable from the initial state. We can speed-up their computation by using backward search to find the former, and forward search to find the latter.

In the rest of the paper we introduce a set of symbolic bidirectional search algorithms to compute centroid and minimum covering states. Experimental results in both existing and novel benchmarks show that these algorithms are faster and scale better than state of the art approaches.

## 2 Preliminaries

### Classical Planning

A $\textsc{sas}^+$ planning task (Bäckström and Nebel 1995) can be defined as a tuple $\Pi = \langle V, I, \mathcal{O}, G \rangle$. $V$ is a finite set of state variables, each associated with a finite domain $D_v$. A partial state $p$ is a function on a subset of variables $V_p \subseteq V$ that assigns each variable $v \in V_p$ a value in its domain, $p[v]$. A state $s$ is a complete assignment to all the variables. With $\mathcal{S}$ we refer to the set of all possible states defined over $V$. We also use partial states to represent conditions on states. A state $s$ satisfies a condition $p$ ($s \models p$) if $s(v) = p(v)$ for all $v \in V_p$. We also identify any partial state $p$ with the set of states that satisfy it: $S_p = \{s \mid s \models p\}$. The state $I \in \mathcal{S}$ is the initial state of the planning task, and $G$ is the goal condition, which defines the set of goal states $\mathcal{S}_G \subseteq \mathcal{S}$.

$\mathcal{O}$ is a set of operators, where an operator is a tuple $o = \langle pre_o, eff_o, c_o \rangle$ of partial variable assignments called preconditions and effects, respectively, and $c_o \mapsto \mathbb{N}_0$ is the nonnegative cost of $o$. An operator $o \in \mathcal{O}$ is applicable in state $s$ iff $pre_o$ is satisfied in $s$, i.e., $s \models pre_o$. Applying operator $o$ in state $s$ results in a state $s[o]$ where $s[o](v) = eff_o(v)$ for all variables $v \in V_{eff_o}$ and $s[o](v) = s(v)$ for all other variables.

A sequence of operators $\pi = \langle o_0, \ldots, o_{n-1} \rangle$ is applicable in a state $s_0$ if there are states $s_1, \ldots, s_n$ such that $o_i$ is applicable in $s_{i-1}$ and $s_i = s_{i-1}[o_i]$ for all $i = 0, \ldots, n$. The resulting state of this application is $s_0[\pi] = s_n$, and $c(\pi) = \sum_{o_i \in \pi} c_{o_i}$ denotes the cost of this sequence of operators. A state $s$ is reachable iff there exist a sequence of operators $\pi$ applicable in $I$ such that $I[\pi] = s$. With $\mathcal{S}_R \subseteq \mathcal{S}$ we refer to the set of all reachable states of the planning task. The solution to a planning task $\Pi$ is a plan, i.e., a sequence of operators $\pi$ such that $I[\pi] \in \mathcal{S}_G$. A plan with minimal cost is optimal.

We denote as $h^*(s, s')$ the optimal cost of reaching state $s'$ from state $s$. If there is no path between the two states, $h^*(s, s') = \infty$. We denote as $g(s) = h^*(I, s)$, and $h_G^*(s) : \mathcal{S} \mapsto \mathbb{N}_0 \cup \{\infty\}$ for a goal $G$ as $\min_{s_G \in \mathcal{S}_G} h^*(s, s_G)$.

### Symbolic Representation with Decision Diagrams

Binary Decision Diagrams (BDDs) (Bryant 1986) are a efficient data-structure to encode Boolean functions $\{0,1\}^n \mapsto \{\top, \bot\}$. We use BDDs to represent sets of states $S \subseteq \mathcal{S}$. This requires to consider some arbirary encoding of the values of the state variables $V$ in binary. To simplify the presentation, we assume without loss of generality that the set of variables $V$ have a binary domain. Each set of states $S$ is represented by a BDD encoding its characteristic function $\mathcal{X}_S : \mathcal{S} \mapsto \{\top, \bot\}$, where $\mathcal{X}_S(s) = \top$ iff $s \in S$. We denote the set of states represented as a BDD as B.

A BDD (Figure 2a) is a directed acyclic graph with a single root node and up to two terminal nodes, $\top$ and $\bot$.

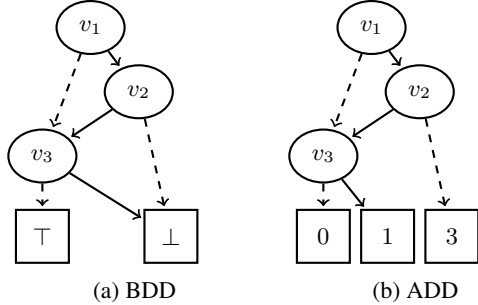

Figure 2: Example of a BDD and an ADD.

Each inner node corresponds to a binary variable $v \in V$, and has two successors depending on whether $s(v) = 0$ (dashed edge), or $s(v) = 1$ (solid edge). The size of a BDD $|\text{B}|$ is simply the number of nodes in its directed acyclic graph. Given a state $s \in \mathcal{S}$, and a BDD B, it can be checked whether $s$ belongs to the set represented by B by a simple top-down traversal, which will always end in $\top$ if $s \in$ B and $\bot$ otherwise. For example, the BDD in Figure 2a has 5 nodes and represents a set with 3 states: $\{v_1 \mapsto 0, v_2 \mapsto 0, v_3 \mapsto 0\}$, $\{v_1 \mapsto 0, v_2 \mapsto 1, v_3 \mapsto 0\}$, and $\{v_1 \mapsto 1, v_2 \mapsto 1, v_3 \mapsto 0\}$. In general, the number of states represented by a BDD can be exponentially larger than its number of nodes. We denote as $\text{B}_p$ the BDD representing a (partial) state $p$, i.e., the set of states $\{s \mid s \models p\}$.

We assume BDDs are *reduced* and *ordered*. BDDs are ordered whenever in any path from the root to the leaves variables are always checked in the same order (though some variables may be skipped). BDDs are reduced whenever (a) there are no irrelevant nodes whose both successor point to the same node; and (b) all nodes are unique (i.e., any equivalent nodes with the same variable and successors are merged). These properties are easy to maintain. Also, they allow to implement efficient operations on BDDs, whose runtime depends on the size of the BDDs and not on how many states they represent (which, again, could be exponentially larger). Specifically, the union ($\cup$) and intersection ($\cap$) of sets of states can be computed as the disjunction (B $\vee$ B$'$) and conjunction (B $\wedge$ B$'$) of their characteristic functions, respectively. The runtime of these operations is $O(|\text{B}||\text{B}'|)$. However, the conjunction/disjunction of $n$ BDDs is worst-case exponential in $n$.

Algebraic Decision Diagrams (ADDs) (Bahar et al. 1997) are similar to BDDs, but have an arbitrary numer of terminal nodes with different discrete values. That is, we can use them to represent the previously defined functions $g$, mapping each state to a numerical value. A schematic representation of an ADD is shown in Figure 2b. The APPLY operation takes an input two ADDs and a binary operation on the terminal values (e.g. $+, -, \times, \div, \max, \min$), and returns a new ADD where the values of its terminal nodes are defined by the result of the binary operations. The runtime is $O(\text{ADD}_1, \text{ADD}_2)$.

ADDs can be converted into BDDs and viceversa (Speck 2022). We will use the BDD$(\text{ADD}, n \in \mathbb{N}_0)$ function to convert ADD into a BDD representing the set of states

$\{s \mid \mathrm{ADD}(s) = n\}$. This can be done in linear time in the size of the ADD by replacing its terminal nodes by $\top$ and $\bot$ and applying the reduction rules in a bottom-up fashion. In the example of Figure 2, the BDD corresponds to $\mathrm{BDD}(\mathrm{ADD}, 0)$.

### Symbolic Bidirectional Search

Symbolic search explores the state space by using the representation of sets of states as BDDs explained above. To do search directly in the symbolic representation, operators are represented as transition relations (TRs). A set of operators $O \subseteq \mathcal{O}$ can be represented as a TR containing the set of all state pairs $(s, s')$ such that $s'$ is reachable from $s$ by applying an operator $o \in O$. For a given set of states B and TR $T$, the image/preimage operator computes all successors/predecessors of B with respect to the operators represented by $T$.

In the forward direction, the search starts with a BDD representing the initial state $\mathrm{B}_I$, and iteratively constructs a set of BDDs with an associated cost $\mathrm{B}_0, \mathrm{B}_1, \mathrm{B}_2$, where $\mathrm{B}_g = \{s \mid h^*(I, s) = g\}$ represents the set of states that can be reached from $I$ with a cost of $g$. Typically, the search terminates whenever the next set of states to be expanded has a non-empty intersection with the set of goal states, also represented as a BDD $\mathrm{B}_G$, meaning that an optimal plan has been found. However, one can also choose to continue the search beyond that point to exhaust the set of reachable states.

On the other hand, symbolic backward search (regression) starts the search from the goal states, and applies the preimage operation until a non-empty intersection with the BDD that represents the initial state $I$ is found. As a result, we obtain BDDs $\mathrm{B}_0, \mathrm{B}_1, \mathrm{B}_2$, where $\mathrm{B}_h = \{s \mid h^*(s, G) = h\}$ represents the set of states that can reach $G$ with a cost of $h$. If the search is exhausted (ignoring $I$), this is equivalent to computing the perfect heuristic $h_G^*$.

The combination of these two searches forms a bidirectional search used by most modern symbolic search planners (Torralba et al. 2017; Speck, Mattmüller, and Nebel 2020). We will use STEP to denote a function that computes the image (preimage) of a forward (backward) search. Finally, we will use FINISHED to denote a boolean function that indicates if the given search has been exhausted, i.e., the image/preimage operation does not generate new non-expanded states.

Perimeter search (Dillenburg and Nelson 1994) consists of searching a perimeter around the goal. This idea has been used in the construction of heuristic functions for explicit-state search using a symbolic backward search, which is interrupted before termination (Kissmann and Edelkamp 2011; Torralba, Linares López, and Borrajo 2018).

Given an unfinished backward search initialized with $G$ where the sets of states $\mathrm{B}_0, \dots, \mathrm{B}_p$ have been generated up to $h = p$, the set of closed states is $closed = \bigvee_{i \in [0,p]} \mathrm{B}_i$, i.e., the set of states for which the real goal distance is known. Then, the perimeter heuristic is $h_G^P(s) := i$ if $s \in \mathrm{B}_i$ and $h_G^P(s) = p + 1$ otherwise. Note that $h_G^P$ is a lower bound on the actual goal distance, i.e., $h_G^P(s) \leq h_G^*(s)$. As sets of states are generated with increasing values of $h$, we are certain that any remaining state will have a goal distance of

at least $p + 1$. We will use CLOSED to denote a function that returns a BDD with the set of states closed by a given search.

## 3 Centroids, Minimum Covering States, and Beyond

In order to compute planning centroids and minimum covering states, we consider the same setting as in previous works (Pozanco et al. 2019; Karpas 2022) using the concept of **Planning task with multiple possible goal conditions** (PMG).

**Definition 1 (PMG)** *A planning task with multiple possible goal conditions is defined as* $\mathcal{P} = \langle V, I, \mathcal{O}, \mathcal{G} \rangle$, *where* $V$, $I$, *and* $\mathcal{O}$ *are defined as in a* SAS$^+$ *planning task, and* $\mathcal{G}$ *is a set of **possible goal conditions**, where each possible goal condition* $G \in \mathcal{G}$ *is defined as in a standard planning task.*

**Definition 2 (Planning centroid states)** *Given a* PMG $\mathcal{P}$, *planning **centroid states** are those reachable states* $s \in \mathcal{S}_R$ *that minimize the sum of costs to the possible goals, i.e.,* $\sum_{G \in \mathcal{G}} h_G^*(s)$.

**Definition 3 (Planning minimum covering states)** *Given a* PMG $\mathcal{P}$, *planning **minimum covering states** are those reachable states* $s \in \mathcal{S}_R$ *that minimize the maximum cost to any of the possible goals, i.e.,* $\max_{G \in \mathcal{G}}, h_G^*(s)$.

We introduce a general definition that encompasses these and other states that optimize a given goal-related function. To do that, we first define goal-related functions.

**Definition 4 (Goal-related function)** *Let* $\mathcal{P}$ *be a* PMG, *and* $h_{\mathcal{G}}^*(s) = (h_{G_1}^*(s), \dots, h_{G_n}^*(s))$, *a vector of optimal costs from* $s$ *to each goal* $G_i \in \mathcal{G}$. *A **goal-related function** is a function* $\phi(h_{\mathcal{G}}^*(s)) \mapsto \mathbb{N}_0 \cup \{\infty\}$.

**Definition 5 (Goals-related states wrt $\phi$)** *Given a* PMG $\mathcal{P}$, *and a goal-related function* $\phi$, *the set of **goal-related states wrt** $\phi$, $S_\phi$, is the set of reachable states that minimize* $\phi$.

This definition generalizes centroids (when $\phi$ is the sum), and minimum covering states (when $\phi$ is the maximum). In principle, one can be interested in any goal-related function. In this paper, we focus in functions that satisfy the following two properties, both of which are satisfied by centroids and minimum covering states.

**Definition 6 (Monotonic goal-related functions)** *A goal-related function* $\phi$ *is **monotonic** iff increasing any of the costs in* $h_{\mathcal{G}}^*(s)$ *does not generate a lower* $\phi$ *value.*

**Definition 7 (Aggregatable goal-related functions)** *A goal-related function* $\phi$ *is **aggregatable** iff its value can be computed by pairwise aggregation of the costs in* $h_{\mathcal{G}}^*(s)$.

Let us clarify these definitions by using a vector of costs $h_{\mathcal{G}}^*(s) = (1, 2, 3)$. In the case of centroids and minimum covering states, $\phi(h_{\mathcal{G}}^*(s))$ can be computed by pairwise aggregation, i.e., $\max(\max(1, 2), 3) = \max(1, 2, 3)$. This is not possible for other goal-related functions such as computing the median of the cost vector. It is also easy to see that when $\phi$ involves minimizing the sum or the maximum, any increase to any component of that vector would entail a worse solution. However, this is not the case for other goal-related functions such as achieving a state with the same cost

to all the goals, i.e., $\phi(h_{\mathcal{G}}^*(s)) = \max h_{\mathcal{G}}^*(s) - \min h_{\mathcal{G}}^*(s) = 0$. In the rest of the paper we will focus on centroids and minimum covering states, which are defined using monotonic and aggregatable goal-related functions, and use these general definitions to simplify notation.

Finally, we are interested in the problem of, given a PMG $\mathcal{P}$, and a function $\phi$, find a state that is a goal-related state wrt $\phi$.

## 4 Computing Goals-Related States with Exhaustive Symbolic Search

Using bi-directional search offers two main advantages over existing approaches: (i) we can perform independent backward searches for each goal; and (ii) we can decouple the search of candidate solutions from the reachability analysis. The first algorithm we propose is $\text{SBD}_e$, an algorithm that exploits these observations by performing (a) exhaustive forward search to find the set of reachable states; and (b) exhaustive backward searches to find the perfect heuristic from each state to each goal. More precisely, $\text{SBD}_e$ starts by performing $|\mathcal{G}|$ independent symbolic backward searches, one from each possible goal $G \in \mathcal{G}$, until all the searches are FINISHED. Each of these backward searches will give us an ADD having at the terminal nodes all the states that can achieve each goal $G \in \mathcal{G}$ with a given cost. We combine all the ADDs into a single backward ADD by using the APPLY operation either with the sum as function, if we are interested in finding centroids, or taking the maximum when computing minimum covering states. We will slightly abuse notation and use $\text{APPLY}(\text{ADDs}, \phi)$ to refer to an operation that perform these pairwise operations over a set of ADDs. Then, the algorithm performs a full forward search from the initial state, yielding the set of reachable states $\mathcal{S}_R$. We update the backward ADD by setting to $\infty$ the value of unreachable states, i.e., those states that do not appear in the CLOSED list of the forward search. Finally, we take the minimum of the backward ADD, and extract a BDD with the states that have that value in the backward ADD. The states in that BDD will be the centroids or minimum covering states, depending on the $\phi$ used in the APPLY operation that combines the backward ADDs. Finally, the algorithm returns any of the states in the BDD as the goal-related state. A plan from $I$ to such state can be easily reconstructed by analyzing the forward search CLOSED list.

This algorithm is symmetric, and we could exchange the forward and backward steps obtaining the same result. It could also be seamlessly adapted to compute goals-related states that optimize any aggregatable goal-related function by just modifying how the ADDs are combined.

## 5 From Exhaustive to Perimeter Search

The main drawback of the exhaustive search is that it performs unnecessary computations by obtaining $h^*$ to each goal from each state in the problem. Some of these computations are not needed, since we are only interested in monotonic goal-related functions, meaning that we can stop the backward searches once a candidate solution is found.

Let us assume a PMG $\mathcal{P}$ with 3 possible goals, where we are interested in computing the minimum covering state. The solution to such task is a single state that is at distance 10 of the further goal, and at distances 4 and 3 of the other two, respectively. In this case, any state with $h^*$ larger than 10 to any goal will not be part of the solution, so it is unnecessary to expand them as the previous exhaustive algorithm does. However, this raises the following question: up to when do we need to keep exploring?

It is clear that we need to explore each search at least up to distances 10, 4, and 3 to each goal, since otherwise we wouldn't know the distance from the solution state to each of the goals. However, that is not sufficient to prove that this is indeed the optimal solution. At this point we know there is no minimum covering state at distance 3 or below, but there could be a solution at distance 5 from each goal. However, we do not need to expand all the backward searches up to distance 10 either. In this case, it would suffice to explore all states at distances 10, 9 and 3, assuming that there is no state at distance 9 or less in the first two searches. Alternatively, we could explore up to 10, 4 and 9, if there is no state at distance 9 or less in the first and third search.

To formalize when we have performed enough search to guarantee that we have found the desired goal-related state, we define the set of candidate states with respect to the perimeter heuristics of all backward searches.

**Definition 8** *Let BW be a set of perimeter searches, and $h_{\mathcal{G}}^P = (h_{G_1}^P, \ldots, h_{G_n}^P)$ a vector of the perimeter heuristics to each goal $G \in \mathcal{G}$. The set of candidate states $B_C$ for some goal-related function $\phi(h_{\mathcal{G}}^*)$ is the set of states that minimizes $\arg\min_{s \in \mathcal{S}} \phi(h_{\mathcal{G}}^P(s))$.*

**Proposition 1** *Let $\phi(h_{\mathcal{G}}^*)$ be a monotone goal-related function. Let BW be a set of perimeter searches with a set of candidates $B_C$. Then, if there exists $s \in B_C$ such that $s \in closed(bw)$ for all $bw \in BW$ and $s \in \mathcal{S}_R$, then $s \in S_\phi$.*

**Proof:** Since $s$ is a candidate, then s has minimum $\phi(h_{\mathcal{G}}^P)$.

Since $s \in closed(bw)$ for all $bw \in$ BW, then $h_G^P(s) = h_G^*(s)$ for all goals $G \in \mathcal{G}$. Therefore $\phi(h_{\mathcal{G}}^P) = \phi(h_{\mathcal{G}}^*)$.

Finally, let $s' \in \mathcal{S}_R$ be any other state. As $h_G^P(s') \leq h_G^*(s')$, by monotonicity of $\phi$, we have that $\phi(h_{\mathcal{G}}^P(s')) \leq \phi(h_{\mathcal{G}}^*(s'))$. So, we conclude that $\phi(h_{\mathcal{G}}^*) = \phi(h_{\mathcal{G}}^P) \leq \phi(h_{\mathcal{G}}^P(s')) \leq \phi(h_{\mathcal{G}}^*(s'))$. As this holds for all $s'$, $s$ is a goal-related state with respect to $\phi$, i.e., $s \in S_\phi$. $\square$

The second algorithm we propose is $\text{SBD}_{bw}$, an algorithm that performs backward search until a solution candidate is found, then running forward search to verify that the candidate solution is reachable. Algorithm 1 shows this process in more detail. At each step (lines 4-18), the algorithm first checks whether there is a candidate solution. This is done by joining all the backward searches in a single ADD through the APPLY operator and the goal-related function $\phi$ (line 4). Then, the algorithm retrieves the lowest value in the terminal nodes of the resulting ADD, and use it to build a BDD with all the states satisfying that value (lines 5-6). After that, the algorithm checks whether those candidate states have been explored by the backward searches (lines 7-8). If that

---
**Algorithm 1:** SBD$_{bw}$

---
**Input:** $\mathcal{P} = \langle V, I, \mathcal{O}, \mathcal{G} \rangle$
**Input:** Goal-related function: $\phi$
**Output:** A goal-related state $s_\phi$

1: fw $\leftarrow$ B$_I$
2: BW $\leftarrow \bigcup_{G \in \mathcal{G}} \{$B$_G\}$
3: **while** true **do**
4:     ADD$_{BW} \leftarrow$ APPLY($\bigcup_{G \in \mathcal{G}} \{$bw$_G$.ADD()$\}, \phi$)  ⎫
5:     value $\leftarrow \min_s$ ADD$_{BW}(s)$                        ⎪
6:     B$_C \leftarrow \{s \mid$ ADD$_{BW}(s) =$ value$\}$            ⎬ Get
7:     B$_{explored} = \bigwedge_{G \in \mathcal{G}}$ bw$_G$.CLOSED()  ⎪ candidates
8:     B$_{expCand} \leftarrow$ B$_{explored} \wedge$ B$_C$           ⎭
9:     **if** B$_{expCand} \neq \varnothing$ **then**
10:        **while** $\neg$fw.FINISHED() **do**           ⎫
11:           B$_\phi \leftarrow$ B$_{expCand} \wedge$ fw.CLOSED()  ⎪
12:           **if** B$_\phi \neq \perp$ **then**           ⎬ Check
13:              **return** any $s_\phi \in$ B$_\phi$       ⎪ reachability
14:           fw.STEP()                                   ⎭
15:     BW $\leftarrow$ bw$_i \in$ BW s.t. $\neg$bw$_i$.FINISHED()  ⎫
        BW $\leftarrow$ bw$_i \in$ BW s.t.                          ⎪
16:                                                                  ⎬ Step
           B$_C \wedge \neg$bw$_i$.CLOSED() $\neq \varnothing$       ⎪ backward
17:     bw_to_advance $\leftarrow$ PICKEASIEST(BW)                   ⎪
18:     bw_to_advance.STEP()                                        ⎭

---

is the case, SBD$_{bw}$ starts progressing the forward search until a non-empty intersection with the candidate solutions is found. The states in that intersection are reachable and conform the set of minimum covering (centroid) states of the task, and SBD$_{bw}$ returns any of the states in that BDD. Otherwise, the algorithm selects which of the backward searches should be progressed (lines 15-18). First, it updates the set of backward searches BW by filtering out those backward searches that (i) have already finished (line 15); and (ii) have already explored the candidate states (line 16). These two filters leave as alternative to progress those backward searches that have not yet explored the states in B$_C$. Then, the algorithm advances the easiest backward search in BW, which we defined as the one that is estimated to generate a lower number of nodes in the next step. This process is repeated until a minimum covering (centroid) is found. This algorithm can be slightly modified to first run forward search to detect all the reachable states, and then perform the backward searches only over these states. We refer to this variation of the algorithm as SBD$_{fw}$.

**Proposition 2** *Algorithm 1 always terminates and returns a goals-related states wrt $\phi$, for any monotone function $\phi$.*

**Proof Sketch:** Termination is guaranteed as at every iteration some unfinished backward search will take a step. The number of steps a backward search is bounded by the number of states in the planning task, so eventually all searches will finish. At that point, the goal distance to all states is known, so the algorithm will return a reachable state with minimum goal-related function value.

Whenever the algorithm terminates, it always returns a goal-related state. Note that all states in B$_\phi$ are candidates (due to line 6), closed in all unfinished backward searches (due to line 7), and reachable (due to line 11). Therefore, all

conditions of Proposition 1 are met.

$\square$

## Lazy Computation using BDDs

In the case of minimum covering states we can slightly modify Algorithm 1 to only use BDDs, which are typically faster to manipulate than ADDs. In this version of the algorithm, we do not select which backward search should be progressed, but perform one step in all of them. This is done by replacing lines 16 to 18 by $\forall_{bw_i \in BW}$bw$_i$.STEP(). Then, we remove lines 4 to 8, and define B$_C$ as the intersection of all the backward searches, i.e., $\bigwedge_{G \in \mathcal{G}}$ bw$_G$. If B$_C \neq \varnothing$, those states represent candidate solutions, and the reachability check (lines 11-14) remains the same. We refer to these variations of the algorithm as SBDD$_{bw}$ and SBDD$_{fw}$, depending on whether we start by finding solution candidates or computing reachable states. Note that we cannot use this variation of the algorithm to compute centroids. In that case we cannot stop the algorithm after finding an intersection at distance 2 of $G_1$ and $G_2$ (sum of 4), since it does not mean that we could not find an intersection at distance 3 from $G_1$ and 0 from $G_2$ (sum of 3).

# 6  Experimental Setting

**Approaches.** We implemented our SBD algorithms on top of Symbolic Fast Downward (Torralba et al. 2017)[1], and compare them against the two other approaches in the literature, GRS and COMP. GRS (Pozanco et al. 2019)[2] uses the Fast Downward planner (Helmert 2006) with the A$^*$ search algorithm (Hart, Nilsson, and Raphael 1968) and the LM-cut heuristic (Helmert and Domshlak 2009) to perform exhaustive search in the original planning task with multiple possible goals $\mathcal{P}$. COMP (Karpas 2022)[3] compiles $\mathcal{P}$ into a standard planning task $\Pi$. We use two planners to solve $\Pi$: the same Fast Downward configuration, as initially proposed by Karpas (2022), COMP$_{fd}$; and the symbolic bi-directional configuration of Symbolic Fast Downward, COMP$_{sbd}$, so we make sure any performance improvement of SBD does not come from using a different planner.

**Benchmarks and Reproducibility.** We used two benchmarks. The first one consists of all domains and problems available at both software repositories[2,3], except for HANOI, which contains tasks with only one possible goal. This gives us four planning domains (BLOCKS, FERRY, GRIPPER, and LOGISTICS), and one grid path-finding domain. The planning domains were adapted from standard IPC benchmarks, and the grid path-finding domains consists of $20 \times 20$ grids with a different percentage of obstacles (5%, 10%, 15%, and 20%). Each domain has 10 problem instances, for a total of 80 problems equally splitted between IPC and grid domains. This benchmark was used by Karpas (2022), and we will refer to it as SMALL, as most of the tasks consists of small planning problems with few possible goals. For example, all BLOCKS instances contain only 5 blocks and 3 goals.

---

[1]https://gitlab.com/atorralba/fast-downward-symbolic
[2]https://github.com/apozanco/GRS_0.1
[3]https://github.com/karpase/grscompilation

| | Centroid | | | | | | Minimum Covering | | | | | | | |
|---|---|---|---|---|---|---|---|---|---|---|---|---|---|---|
| Domain | GRS | COMP$_{fd}$ | COMP$_{sbd}$ | SBD$_e$ | SBD$_{bw}$ | SBD$_{fw}$ | GRS | COMP$_{fd}$ | COMP$_{sbd}$ | SBD$_e$ | SBD$_{bw}$ | SBD$_{fw}$ | SBDD$_{bw}$ | SBDD$_{fw}$ |
| BLOCKS (10) | **10** | **10** | **10** | **10** | **10** | **10** | **10** | **10** | **10** | **10** | **10** | **10** | **10** | **10** |
| FERRY (10) | 0 | 9 | 2 | **10** | **10** | **10** | 0 | 7 | 6 | **10** | **10** | **10** | **10** | **10** |
| GRIPPER (10) | 1 | **10** | **10** | **10** | **10** | **10** | 2 | **10** | 8 | **10** | **10** | **10** | **10** | **10** |
| LOGISTICS (10) | 5 | **10** | 4 | **10** | **10** | **10** | 7 | **10** | **10** | **10** | **10** | **10** | **10** | **10** |
| GRID (40) | 18 | 39 | 1 | 1 | 2 | **40** | 18 | 1 | 1 | 1 | 3 | **40** | 3 | **40** |
| SMALL (80) | 34 | 78 | 27 | 41 | 42 | **80** | 37 | 38 | 35 | 41 | 43 | **80** | 43 | **80** |
| BLOCKS (80) | 2 | 30 | 10 | 45 | 58 | **60** | 2 | 9 | 10 | 45 | **67** | 59 | 62 | 60 |
| FERRY (80) | 0 | 46 | 27 | **80** | **80** | **80** | 0 | 39 | 31 | **80** | **80** | **80** | **80** | **80** |
| GRIPPER (80) | 2 | 74 | 32 | 46 | 70 | **75** | 2 | 55 | 25 | 46 | 71 | 75 | 72 | **77** |
| LOGISTICS (80) | 3 | 57 | 32 | 64 | **70** | 62 | 3 | 27 | 26 | 64 | **72** | 64 | **72** | 64 |
| GRID (80) | 15 | 50 | 16 | 0 | 1 | **58** | 17 | 19 | 15 | 0 | 7 | **60** | 6 | **60** |
| LARGE (400) | 22 | 257 | 117 | 235 | 279 | **335** | 24 | 149 | 107 | 235 | 297 | 338 | 292 | **341** |
| TOTAL (480) | 56 | 335 | 144 | 276 | 321 | **415** | 61 | 187 | 142 | 276 | 340 | 418 | 335 | **421** |

Table 1: Coverage of the approaches in computing centroid and minimum covering. Bold figures indicate best performance.

The second benchmark, which we will refer to as LARGE, consists of novel $\mathcal{P}$ tasks in the same five domains, where we generated 80 tasks of increasing difficulty. In BLOCKS, we generated these instances by creating random configurations of 6, 8, 10 or 12 blocks, and having 2, 4, 8 or 16 possible goals (words) to be formed by stacking the blocks. In GRID, we generated $10\times10$, $20\times20$, $40\times40$ and $80\times80$ grids, with 2, 4, 8 or 16 possible goals (agent's locations). For FERRY, GRIPPER and LOGISTICS, we selected the 20 first problems available at the Planning Domains repository[4], and created 4 different instances for each problem by generating between 2 and 5 possible goals. These goals are sets of cars/balls/packages being delivered at different locations/rooms/cities.

Experiments were run on an Intel Xeon E5-2666 v3 CPU @ 2.90GHz x 8 processors with a 8GB memory bound and a time limit of 1800s. Code and benchmarks will be made available upon paper acceptance.

# 7 Results

In this section we present the results of our evaluation, where we aim to investigate how our algorithms compare to existing approaches, in terms of coverage and execution time.

## Coverage Analysis

Table 1 presents the results of our first analysis, where we focus on studying the coverage ($C$) of each approach, i.e., the number of problems they solve.

Regarding centroids (left side of the table), COMP$_{fd}$ solves more than twice the tasks solved by GRS in the small instances, as previously reported by Karpas (2022). This performance difference is emphasized in the larger tasks, where GRS solves only 22 tasks compared to the 257 for which COMP$_{fd}$ can compute a centroid state. Solving the compiled tasks using a symbolic planner (COMP$_{sbd}$) offers a worse performance across all domains, highlighting the fact that the reformulated task does not suit symbolic planners particularly well. Our baseline approach that performs exhaustive search (SBD$_e$) achieves results that are on par with COMP$_{fd}$

[4]https://github.com/AI-Planning/classical-domains

in all small instances except for those from the GRID domain, where SBD$_e$ can only solve 1 out of 40 problems. This is because this arguably ill-defined domain uses a redundant (free ?c) predicate to denote that a given cell is free and the agent can move to it. Having this predicate turns GRID tasks into challenging for approaches using backward search, since the set of reachable states is orders of magnitude smaller than the set of states. For example, assuming a $20 \times 20$ grid without obstacles, the problem will have only 400 reachable states, but more than $2^{400}$ total states that backward searches could potentially consider. This is also the reason why SBD$_{bw}$ obtains similar results, although it outperforms the exhaustive algorithm in larger tasks, being able to solve 44 tasks more. In fact, if we leave GRID aside, SBD$_{bw}$ would be the best performing algorithm, solving 1 task more than SBD$_{fw}$, the winner across the benchmark with a coverage of 415. By first computing the reachable states and using that information to prune the backward searches, SBD$_{fw}$ is able to solve most of the GRID tasks, as well as showing a good performance in the rest of domains.

Similar conclusions can be drawn from the minimum covering states results (right side of Table 1). In this case, SBD approaches outperform the compilation approach by even larger margins, with the best performing SBD variant solving 421 tasks compared to the 187 solved by COMP$_{fd}$. This is because the compilation to compute minimum covering states is more involved than the centroids one, since it needs to discretize numerical variables. The performance of the SBDD algorithms is very similar to their SBD counterparts.

## Runtime Analysis

Coverage results show that SBD approaches tend to outperform COMP approaches. Next, we conduct an analysis of execution time to test this further and study in more detail the differences between all the approaches. A summary of these results is shown in Figure 3, where we represent the execution time in log scale of pairs of approaches. Each point in the plot corresponds to a problem of our joint benchmark (480 tasks), with its color indicating the domain it belongs to. Points above the diagonal indicate that the approach in the $x$ axis is faster than the approach in the $y$ axis.

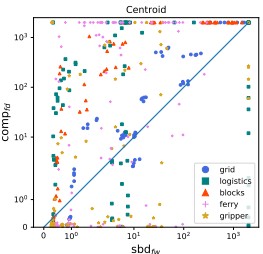

(a) Best SBD and COMP approaches when computing centroids.

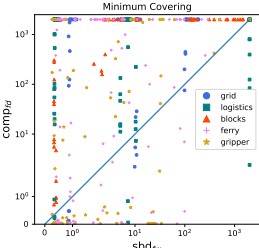

(b) Best SBD and COMP approaches when computing minimum covering states.

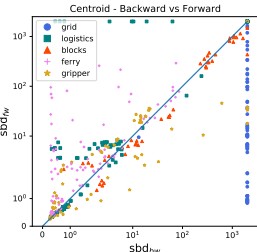

(c) $\text{SBD}_{bw}$ vs $\text{SBD}_{fw}$ when computing centroids.

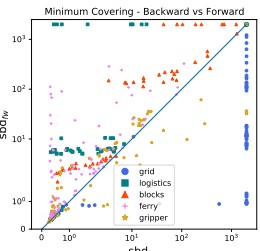

(d) $\text{SBD}_{bw}$ vs $\text{SBD}_{fw}$ when computing minimum covering states.

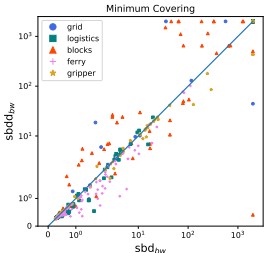

(e) $\text{SBD}_{bw}$ vs $\text{SBDD}_{bw}$ when computing minimum covering states.

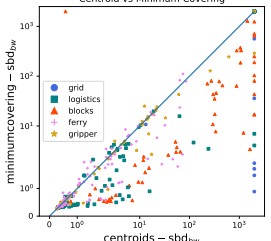

(f) $\text{SBD}_{bw}$ when computing centroids vs minimum covering states.

Figure 3: Pairwise comparisons of execution time.

**SBD VS COMP.** First, we want to compare the execution time of the best variants of each approach. These results are shown in Figures 3a and 3b, where we compare the execution time of $\text{SBD}_{fw}$ ($x$ axis) and $\text{COMP}_{fd}$ ($y$ axis). In the case of centroids (Figure 3a), $57\%$ of the tasks were solved faster by $\text{SBD}_{fw}$ than by the compilation, i.e., $57\%$ of the points fall above the diagonal line in the plot. We can observe some trends across domains. For example, $\text{SBD}_{fw}$ is consistently faster than $\text{COMP}_{fd}$ in BLOCKS and GRID instances, while the results in the other domains depend more on the task at hand. There is a cluster of simple problems that $\text{COMP}_{fd}$ can solve in less than a second, while $\text{SBD}_{fw}$ requires up to 1 second to solve them. Symbolic Fast Downward initializes some data structures before starting the search process, which slightly delays SBD algorithms in some simple tasks. Following the coverage trend, the execution time difference between both approaches increases in the case of minimum covering states (Figure 3b), with $67\%$ of the tasks being solved faster by $\text{SBD}_{fw}$.

**Backward vs Forward.** Forward SBD variants obtained higher coverage scores mainly due to GRID, but the forward variants were able to solve more problems in domains such as LOGISTICS and BLOCKS. Figures 3c and 3d compare the execution time of $\text{SBD}_{bw}$ and $\text{SBD}_{fw}$ when computing centroids and minimum covering states, respectively. As we can see, $\text{SBD}_{bw}$ is faster than $\text{SBD}_{fw}$ when finding both states in most domains and tasks except for GRID. This was expected, as $\text{SBD}_{bw}$ does not need to compute all the set of reachable states as $\text{SBD}_{fw}$ does, which might be demanding in some problems. In particular, starting the search backwards until a minimum covering state candidate is found is faster in $64\%$

of the tasks. This percentage raises up to a $86\%$ if we do not consider GRID, re-emphasizing $\text{SBD}_{bw}$ as the go-to algorithm for many IPC domains such as BLOCKS where (i) most of the states in the planning task are reachable; and (ii) the number of these states is large.

**SBD VS SBDD.** The coverage difference between the SBD variants, which use ADDs and BDDs and progress only one backward search at a time, and the SBDD variants, which only use BDDs and progress multiple backward searches simultaneously, was negligible. Figure 3e compares the execution time of their backward versions when computing minimum covering states. As we can see, both approaches have very similar execution times. These similar runtimes are explained by two counteracting factors. On the one hand, SBDD variants should be faster, as they do not need to build and reason over ADDs. On the other hand, SBD variants should be faster, as they typically require to advance a lower number of backward searches.

**Centroids vs Minimum Covering States.** Coverage figures suggest that while finding minimum covering states is much harder for the compilation approaches, it is as difficult as computing centroids for our SBD approaches. Figure 3f compares the execution time of $\text{SBD}_{bw}$ to find each of the states in the $480$ tasks. The results clearly indicate that computing minimum covering states is faster than computing centroids, with only $17\%$ of problems where the opposite is true. Combining the ADDs of the backward searches by taking their maximum generates a smaller joint ADD than when these ADDs are combined by taking their sum, as the number of possible terminal node values is larger in the latter case. If we empirically analyze these differences for the ex-

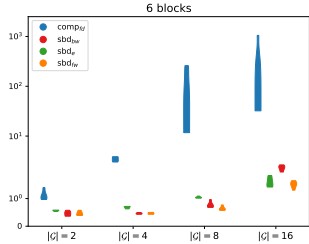 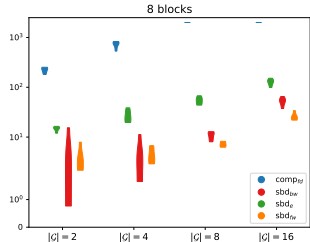 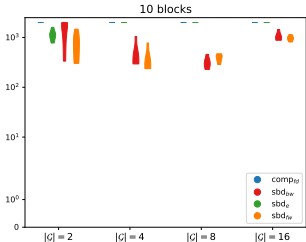

Figure 4: Execution time (log scale) distribution of each approach when computing minimum covering states in BLOCKS tasks. Each graph shows a set of violinplots representing the execution time distribution ($y$ axis) of each approach (different colors), as we fix the problem size (6, 8 and 12 blocks) and vary the number of possible goals ($x$ axis).

haustive algorithm $\text{SBD}_e$, which needs to execute the same number of step operations for both states, we observe that the ADD operations require $16.1 \pm 77.6$ seconds in the case of centroids, versus the $12.4 \pm 57.1$ required in the case of minimum covering states.

## Scalability Analysis

Finally, we study the scalability of the different approaches. Figure 4 shows the execution time of $\text{COMP}_{fd}$ and the three SBD variants when computing minimum covering states in BLOCKS, where we generated problems with increasing complexity by varying the number of blocks (size of the problem) and the number of words to build (number of possible goals). Each graph shows a set of violinplots representing the execution time distribution ($y$ axis) of each approach (different colors), as we fix the problem size and vary the number of possible goals ($x$ axis). We do not report results in problems with 12 blocks, since none of the approaches is able to solve any of the tasks.

As we can see, our SBD approaches scale much better than the compilation ones, which cannot solve any task with 10 blocks (right plot) or with more than 4 goals in problems with 8 blocks. As expected, the exhaustive approach tends to be slower and scales worse than the backward and forward variants. This difference is not clear for small instances (left plot), mainly due to the low execution times and the limited sample of 5 problems per each combination of possible goals and number of blocks. The results indicate that problem size is the main execution time driver for our approaches, while the number of possible goals is the main factor in $\text{COMP}_{fd}$ execution time. Similar results were obtained in the case of GRID, the other domain with controlled problems, with the caveat that, as we already discussed, the backward searches cannot solve many of these tasks.

## 8 Conclusions and Future Work

In this paper we have presented SBD, a family of symbolic bidirectional search algorithms to compute centroids and minimum covering states. Experimental results in existing and novel benchmarks show that our algorithms outperform current approaches both in coverage and execution time. These results establish our algorithms as a new state-of-the-art technique for this task. The decision of which SBD algo-

rithm should be used greatly depends on the task at hand. Forward alternatives are the best option when the number of reachable states is small compared to all states in the planning task, as it can prune many states and simplify the backward searches. Backward alternatives are the best choice in the rest of cases.

This work paves the path for many interesting research avenues, that we group into two different areas: performance and generalization. We would like to further improve SBD performance in three directions. First, SBD advances the backward search that is estimated to generate a lower number of nodes in the next step. In future work, we would like to explore different heuristics to make this decision. Second, we would like to incorporate the $h^2$ preprocessor (Alcázar and Torralba 2015) to our planners, so we can increase the coverage in the variants that prioritize backward search, which tend to be faster. Third, we would like to estimate reachability with heuristics without conducting the forward search. This remains an open question, as it would require computing heuristics with respect to a set of goal states (our candidate set) represented as a BDD. Currently, this is only possible for admissible heuristics that are derived with symbolic search (Torralba et al. 2016; Torralba, Linares López, and Borrajo 2018), but these are not competitive for satisficing planning. Finally, in this paper we have focused on two goal-related states that have already been defined and proved useful in the literature. In future work we would like to generalize SBD to consider other goal-related states, such as those that are at the same distance from all the goals, or those that are as far as possible from the goals.

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
