# OpenReview forum: "Computing Planning Centroids and Minimum Covering States using Symbolic Bidirectional Search"
_icaps-conference.org/ICAPS/2024/Conference — ICAPS 2024_

### Official Review · Reviewer_KCye · 2023-12-26

**Significance And Importance:** 2
**Soundness:** 3
**Novelty:** 2
**Clarity:** 4
**Overall Evaluation:** 2
**Confidence:** 3

**Weaknesses:**

2: No major or minor weaknesses.

**Contributions Of The Paper:**

The paper introduces the use of symbolic bidirectional search for finding states that are relevant to a set of possible goal states. These relevant states are centroids, which minimize the average / total distance to the set of possible goals, and minimum covering states, which minimize the maximum possible distance to a goal.

While previous approaches use exhaustive search or a multi agent compilation to find these states, the proposed approach instead uses symbolic search. This allows forward and backward searches to be performed iteratively until they meet in the middle. Several variations of the approach are proposed that are different in the order in which they conduct searches and their use of ADDs, in which leaf nodes are integers, vs. BDDs, in which leaf nodes are boolean values. This approach is shown to strongly outperform the existing approaches that the paper uses as a benchmark.

**Ethical Considerations:**

(1) Not Applicable: The paper does not have any ethical considerations to address

**Nomination For Best Paper:**

No

**Questions For Authors:**

- Additional knowledge, such as probabilities associated with each goal, or different per goal utility values, might complicate this problem. Would your methods generalize to these cases?

- The scalability results are reported mostly in terms of runtime in the results section. Do you have any estimates of how much exploration your approach saves in terms of the proportion of the state space is explored? This might be interesting to add.

- Consider a blocksworld problem in which on (A, B) is *not* part of the goal. Then for any state in which on (A, B) is true, there should be an equivalent state that is just as good in terms of the goals-related function phi in which on (A, B) is not true. This could be thought of as a form of inference or symmetry pruning. Have you investigated such approaches?

minor:

"The runtime is O(ADD1, ADD2)." is there a missing function here, e.g. max?


typos, etc:

there exist*s* a sequence

arbi*t*rary

to refer to an operation that perform*s*

splitted --> split

"These two filters leave as alternative to progress those backward searches that have not yet explored the states in BC." this sentence is a bit confusing, suggest rewriting.

"The states in that intersection are reachable and conform the set of minimum covering (centroid) states of the task" 'conform' seems like the wrong word here, 'contain' maybe?

"returns a goals-related states" remove 'a', or states --> state

**Reproducibility:**

4: Authors promise to release code and domains (whichever apply).

**Strengths Of The Paper:**

The paper proposes a simple idea which is clearly laid out and well illustrated with examples. The writing is clear. The evaluation is fairly exhaustive and makes sense, and the approach is shown to conclusively outperform existing methods.

**Weaknesses Of The Paper:**

The proposed technique is a fairly straightforward application of symbolic search to the problem, but given that this has not previously been tried I don't consider this a weakness of the paper. I don't think the paper has any significant weaknesses.

---

> ### Author Rebuttal · Authors · 2024-01-27
>
> Thank you very much for your comments, we will clarify all those details in the paper.
>
> Q1) Yes, as mentioned in the response to Reviewer dBEn, we were considering further possibilities for operator functions. The algorithm directly applies to goals with weights/utilities/probabilities, since one can simply multiply each goal-distance by a constant before adding them up (in centroids), or taking the maximum.
>
> Q2) That's difficult to measure. In explicit-state search, the amount of explored states is representative of the search effort. In symbolic backward search, however, the BDD representing the set of all goal states can easily be constructed. And yet, it can represent exponentially many states (e.g. already have half of the states in the state space without any search effort).
>
> And against the compilation approach, the search happens in a different state space, so the comparison in terms of states also does not make sense there.
>
> Q3) For on(A,B) to be irrelevant, not only it should not be a goal, but also the goal-distance for each goal should be independent of whether on(A,B) is true or not. It's likely that some pruning techniques for classical planning could be used if they apply for all goals (e.g. eliminating everything that is irrelevant for all goals, or applying some symmetries that hold for all goals). But in the case of blocksworld is not that simple, as on(A,B) can influence the goal distance of any goal. For example, if we have clear(C) as a goal, and we are in some state in which on(B,C) is true, the goal distance may be affected by on(A,B).
>
> Q4) Yes, the runtime should be the product of the sizes of A and B.
>
> We will fix the typos and improve the writing in the final version of the paper.

---

### Official Review · Reviewer_dBEn · 2024-01-18

**Significance And Importance:** 2
**Soundness:** 3
**Novelty:** 3
**Clarity:** 4
**Overall Evaluation:** 2
**Confidence:** 4

**Weaknesses:**

1: Minor weaknesses that are easily fixable.

**Contributions Of The Paper:**

This paper describes a symbolic bidirectional search approach to solving two related problems: centroid goal coverage and minimum covering states, both of which find a state that ‘covers’ a set of auxiliary goal states in slightly different ways.  The authors show how states can be represented using decision diagrams (ADDs and BDDs), and operations on those states can be used to drive symbolic search.  Multiple algorithms are presented using these techniques.  These algorithms are shown to improve on prior state of the art algorithms on IPC benchmark problems.  Runtime comparisons are performed on the new algorithms, showing which is best.  In particular, one of the two problems (minimum covering sets) is easier than the other (centroid), backward search is faster than forward search on most domains (but one domain favors forward search), and switching out ADDs for BDDs showed little benefit.

**Ethical Considerations:**

(1) Not Applicable: The paper does not have any ethical considerations to address

**Nomination For Best Paper:**

No

**Questions For Authors:**

One way of thinking about this line of work is that the centroid or covering goals essentially create a deterministic/classical problem in lieu of an expected value problem like an MDP or contingent planning problem.  In the forest fire example, we don’t know which fires will break out, so we want to position ourselves to handle them if and when they do.  How would one compare the problem formulations and solutions from these two disparate perspectives?

Are there ever problems in this setting that have no solution?  What would such a problem look like?  If any of the goals G is unreachable that’s enough for a problem to have no solution; if all the goals are reachable, then there has to be some state that solves the problems by definition (I think)

For future work, what if the goals are temporally scoped in some way, such as, take a picture of location A at time t1 or location B at time t2?  The problems are well defined because the definitions of ceontroid and minimum cover only involve cost, and the problems are all posed in terms of reachability, so temporal constraints limit state reachability. However, I could see the resulting minimum covering state being meaningless, i.e. it’s impossible to actually get to either location at the right time from any state other than the initial state.  Example: you can get to A by t1 from where you are, but you have to go the opposite direction to get to B by t2; the only time/place pair in common is the initial state.  So problems may have no solution, or the only solution may be 'trivial'.  What future work would be needed to extend the problems and algorithms?  Will BDDs work?  What do you do if they don’t?

**Reproducibility:**

5: Code and domains (whichever apply) are already publicly available

**Strengths Of The Paper:**

The problem is an interesting take on planning; it can be thought of as a flavor of 'conformant' planning inasmuch as the centroid or minimum cover is 'preparing for the worst'.  The paper addresses algorithmic shortcomings of prior approaches to solving the problem.  The use of BDDs and forward-backward search and the different approaches are interesting.  The paper is technical but well written.  The empirical results look good and the analysis is well done.

**Weaknesses Of The Paper:**

The paper contains no siginficant weaknesses but I had a number of specific comments.

SPECIFIC COMMENTS
p. 1: minimum covering state: does this state minimize the maximum shortest path between the state and any of the goals?  That seems to be the case (because another interpretation of this statement is the worst path one could take which introduces looping plans which could be infinite cost, which probably isn’t what is intended.)

p. 2: “This requires to consider some arbirary encoding of the values of the state variables V in binary.” should be written
“This requires some arbirary encoding of the values of the state variables V in binary.”

p. 2: “Also, they allow to implement efficient operations…” should be written “Also, they allow implementation of efficient operations…”

p. 2: “The runtime is O(ADD1, ADD2).”  What does this mean?  could this be missing a ‘max’ function?

p. 3: definition of monotone.  It may be worth emphasizing in the definition that increasing any of the single costs *alone* can’t decrease the h* value, but (as is the case with minimum covering states) that does not preclude a lower h* value arising from an increase in one cost and decreases in another cost.

p. 4: the narrative introduction to SBD_e and the more detailed description don’t quite match.  If you just reorder the narrative so that forward reachability is done first, then backward search from each goal G and combining the resulting ADD’s then you’re fine

p. 4 some sentences have some awkward repetition (Finally….Finally…)

p. 4: I am having some trouble with the discussion leading up to defn 8 / proposition 1.  Are we provided with *a* state that has distances 10,4,3 to the three goals?  Or is the argument that this state is the solution to this problem that search does not yet know is a solution?

p. 4: Definition 8 may not be written quite right from an editorial point of view.  Since you want states, mathematically speaking, arg_min returns those states minimizing your heuristic, so I think you can just drop ‘that minimizes’ from the textual definition and you’re fine

p. 5:”The states in that intersection are reachable and con- form the set of minimum covering (centroid) states of the task…” should be written “The states in that intersection are reachable and contain the set of minimum covering (centroid) states of the task…”

p. 6: “Having this predicate turns G R I D tasks into challenging for approaches using backward search…” should be written “Having this predicate turns G R I D tasks into challenging ones for approaches using backward search…”

p. 7: “ Forward S B D variants obtained higher coverage scores mainly due to GRID, but the forward variants were able to solve more problems in domains such as LOGISTICS and BLOCKS.”

 I think you meant to say “…the backward variants were able to solve more problems in domains such as LOGISTICS and BLOCKS.”  This is what the results table indicates.

---

> ### Author Rebuttal · Authors · 2024-01-27
>
> Thank you very much for your comments, we will clarify all those details in the paper.
>
> Q1) How would one compare the problem formulations and solutions from these two disparate perspectives?
>
> As the reviewer mentions, one could think about our suggested states as the best initial states from which to start a planning/MDP problem. In the case of a planning problem, it would be the state from which you could get on average (centroid), the shortest path to achieve any of the potential goals that could appear. In the case of a (discounted) MDP, it would be the state that would allow you to get a higher reward whenever one of these goals appear. Note that our states are useful when no goal has appeared yet: whenever one of the goals is achieved, the centroid/minimum covering state, towards the agent should move to, will need to be re-achieved.
>
> Q2) Are there ever problems in this setting that have no solution?
>
> As Reviewer SbYu points out, we assumed for simplicity that all goals are reachable. Given the criteria as defined, if any goal is unreachable, then all states are equally good (value is infinity) so one can simply return the initial state. We will make this more precise and clear in Algorithm 1 and Proposition 2.
>
> Q3) Temporally-extended goals:
>
> That's an interesting future direction that we had not considered. Some of the simpler variants would be directly possible with the current framework. We were already considering what other goal-related functions could be interesting. If one encodes time in the cost, and each goal has a maximum deadline (cost bound), it is possible to define a function that simply counts the amount of goals whose distance is beyond their bound. f(h1,...hk) = sum_i [hi > Ci] where [hi > Ci] is 1 if the condition is met and 0 otherwise. Minimizing such function corresponds to the state from which more goals would be achievable.
> Considering more complex temporal/concurrent actions would probably require integrating temporal planning techniques.
>
> p.1) Yes, minimum covering states minimize the maximum shortest path between the state and any of the goals. We will clarify it in.
>
> p.3) We will clarify that we refer to single costs being increased.
>
> p.4) We will be consistent when introducing and describing SBD_e. Regarding the discussion leading to Def.8, the argument is that this state is the solution to the problem. We will clarify it.
>
> We will fix the typos and improve the writing in the final version of the paper.

---

### Official Review · Reviewer_SbYu · 2024-01-19

**Significance And Importance:** 3
**Soundness:** 2
**Novelty:** 3
**Clarity:** 2
**Overall Evaluation:** 1
**Confidence:** 4

**Weaknesses:**

0: Minor weaknesses requiring some work to be addressed for the paper to be accepted.

**Contributions Of The Paper:**

The paper suggests a new way of computing states that satisfy some condition on their distances to a set of given goal states, such as minimizing the average or maximal distance to a goal. Compared to previous approaches, the authors suggest using symbolic bidirectional search. One direction computes the desired function backwards from the goals while the other checks for reachability in the forward direction. Experiments clearly show that the new methods outperform the state of the art.

**Ethical Considerations:**

(1) Not Applicable: The paper does not have any ethical considerations to address

**Nomination For Best Paper:**

No

**Questions For Authors:**

1. How does SBD_e update the backward ADD (see Section "Weaknesses" for the part that was unclear to me)?
2. Why is the lazy computation with BDDs correct for computing minimum covering states?

Response to Rebuttal
=================
Thank you for your response. I still believe the experimental section could be compressed a lot (specifically points (1) and (2) in your response) but I'm happy with all suggested changes.

**Reproducibility:**

4: Authors promise to release code and domains (whichever apply).

**Strengths Of The Paper:**

The contribution is a good fit for ICAPS and the algorithm presented in the paper is novel. The experimental results show a clear advantage over existing methods and the theory is convincing.

**Weaknesses Of The Paper:**

My main issue with the paper is clarity. While the writing is generally very good, some technical aspects are unclear (concrete points listed in the section "Clarity" below). In these cases, the paper is understandable on an intuitive level but is missing details to understand the formal aspects. For the most part, I believe this to be easily fixable by extending the theoretical part of the paper (the intuitive explanations currently in the paper are useful, so theoretical details should be added to them not replace them).

In addition, the experimental design is not that informative. The main important questions are answered by the discussion of coverage but the remaining sections (in particular, "Runtime Analysis") do not add much additional insight. Configurations that have higher coverage than other configurations are show to run faster. This is not that surprising as the higher coverage could only be caused by lower run time or memory usage. I would suggest to cut most or all of this section and use the extra space for more details in the theoretical part of the paper.

There are two minor soundness issues in the correctness proof (Proposition 2) but these are mainly corner cases:
* Algorithm 1 doesn't specify what happens if BW is empty after filtering
* Proposition 2 (and maybe Algorithm 1) seems to assume that all goals are reachable and that a goal-related state wrt. phi exists.


Clarity
=======

In several places, the paper uses BDDs and ADDs in a manner that intuitively describes the operation without being clear on how that operation is performed. For example, Section 4 states: "Each of these backwards searches will give us an ADD having at the terminal nodes all the states ..." but formally, ADDs do not have states at their terminal nodes but numbers. Shortly after: "We update the backward ADD by setting to $\infty$ the value of unreachable states". It is still not clear to me how this is done. Would this iterate over unreachable states and do something for each one? How can you set the value of a state in an ADD? My guess is that this is some ADD "apply" operation with the ADD and some ADD representing the closed states but this is not explicit in the text.

The end of Section 4 mentions that the algorithm is symmetric but it isn't: The forward search is a single search that starts in one state, while there are several backward searches that start from multiple goals. Backward searches are represented with ADDs, the forward search is unspecified (it could be an explicit search). It is not clear at all that forward and backward step are exchangeable.

The discussion above Definition 8 is unclear as well. For one, the text doesn't justify statements like "we could explore up to 10, 4 and 9" but just states them as facts. Reading Proposition 1 afterwards does not help, because it doesn't mention g values, only closed states, so it doesn't directly apply to the text above.

In the pseudo code, lines 15 and 16 are unclear: they look like they would assign one element out of BW to BW, turning BW from a set of searches to a search. It would help to change $bw_i \in BW s.t. ...$ to $\{bw_i \in BW \mid ...\}$. Another issue is that BW is not a local variable, i.e., the global list of all searches is modified and some elements are removed from it forever. After some thought I convinced myself that this is fine but it makes the algorithm much harder to understand. If you would instead use a local variable here and filter the list of searches only for use in line 17, this would be easier.

Finally, the description of the special case using BDDs instead of ADDs is unclear. When doing just the modifications to the pseudo code described in the text, line 9 uses a variable B_expCand that was never set (because lines 7 and 8 were removed). Then, the conjunction of bw_G is not consistent with using bw_G.CLOSED() in other places (I assume this should also use the closed list of the search). Pseudo code for the modification and an argument for correctness of the algorithm would help here.

Minor comments
==============
* Introduction: "arrive" seems like the wrong verb for a fire. How about "break out"?
* Figures 1, 3, and 4 are hard to parse in a black-and-white printout. Maybe change "(blue)" to "(C, blue)" and use different colors in Figures 3 and 4.
* The symbol \mapsto is misused in different ways: it should only be used as part of a function definition ($s = {v_1 \mapsto 1}$) but not as defining the functions type (use $\chi: S \to {T, F}$ instead of $\chi: S \mapsto {T, F}$) or for individual elements (use $c_o \in \mathbb{N}_0$ instead of $c_o \mapsto \mathbb{N}_0$). In particular in Definition 4, $\phi(h(s))$ is an element of the image of the function, so $\phi(h(s)) \mapsto \mathbb{N}_0 \cup {\infty}$ doesn't make sense. This is also mixing concrete elements
  ($\phi(h(s))$) and the type of those elements ($\mathbb{N}_0 \cup {\infty}$).
  A fix would be to just specify the type as
  $\phi: (\mathbb{N}_0 \cup {\infty})^n \to (\mathbb{N}_0 \cup {\infty})$.
* "We denote the set of states ... as B": it is not clear if B should refer to the set or the BDD. Also, later you introduce B_p for the same concept.
* "nodes whose both successor point to the same node": nodes _in_which_ both successor_s_ point to the same node
* "the previously defined functions g": singular
* "takes an input two ADDs": as input?
* "The runtime is O(ADD_1, ADD_2)": O([|ADD_1|] * [|ADD_2|])
* "viceversa": two words
* Definition 7 defines one hand-wavy term (aggregatable) by another (pairwise aggregation). As this is a formal definition, the second one should be more precise.
* "the problem of, given": grammar. Maybe "The problem of finding ..., given ..."
* "The solution to such _a_ task"
* "assuming there is no _reachable_ state at distance 9"
* "turns GRID tasks into challenging" incomplete sentence
* Figure 4: "6, 8, and _10_ blocks"

---

> ### Author Rebuttal · Authors · 2024-01-27
>
> Thank you very much for your comments, we will clarify all those details in the paper.
>
> Q1) Indeed, this can be done using apply. Specifically, let A be an ADD and B a BDD representing the unreachable states. We first obtain an ADD A_B, which replaces the terminals of the BDD B true->infinity and false->0. And then, use the apply operation apply (max,A,A_B), which assigns each state to the maximum value assigned by either A or A_B. The runtime does not depend on the number of represented states, it is bounded by the product of the sizes of A and A_B (whose size is equal to that of the BDD B).
>
> Q2) Algorithm 1 (SBD) has two main steps: computing a candidate solution and checking if such candidate is reachable. The candidate solutions are computed by advancing one of the backward searches at each iteration, and checking if a non-empty intersection exists among all the backward searches. In the lazy version (SBDD), each iteration progresses all the backward searches instead of only one. We can do this for minimum covering states (MCS), since in this case we are interested in minimizing the max distance to *any* of the goals. Let us assume we have two goals G1 and G2, and we find the first non-empty intersection at distance 2 from both goals. In the case of MCS this is a valid candidate, since any other combination of distances (1 from G1 and 2 from G2) would have the same max distance. However, this is harder for centroids, since we cannot stop the algorithm after finding the first non-empty intersection: in this case 1 from G1 and 2 from G2 would be a better solution.
>
> In our Runtime Analysis we intended to complement the coverage analysis (1) show how much faster our approaches are compared to SoTA; (2) show in more detail how the runtime scales wrt. the number of goals and size of the problem; and (3) understand whether computing centroids is harder than minimum covering states.
> However, we will make sure that the theory part is well-explained, and summarize/compact some of these experiments if needed for space reasons.
>
> By symmetric we meant that in the case of SBD_e you could exchange the order (bw  first, fw first) and you would get the exact same results (state returned) and algorithm's performance (expanded nodes and runtime)
>
> BW contains a set of searches (line 2). In lines 15 and 16 we are updating this set filtering out those bw searches that (i) have already finished (line 15); and (ii) have already explored the candidate states (line 16).

---

### Meta-Review · Area_Chair_Lckj · 2024-01-31

**Recommendation:** Accept (Oral)
**Confidence:** 5

**Metareview:**

All reviewers liked the paper and agreed that its contribution is worth publishing. Some concerns, e.g., some clarity issues, were raised but the reviewers all agreed that they believe that these issues will be resolved in the revised version. With therefore recommend accepting this paper.

**Ethical Considerations:**

(1) Not Applicable: The paper does not have any ethical considerations to address